# The Prognostic Value of Pulmonary Hypertension in Patients with Mitral Regurgitation Undergoing Mitral Valve Transcatheter Edge-to-Edge Repair: A Systematic Review and Meta-Analysis

**DOI:** 10.3390/diagnostics15070852

**Published:** 2025-03-27

**Authors:** Yanbiao Liao, Junli Li, Mao Chen

**Affiliations:** 1Department of Cardiology, West China Hospital, Sichuan University, Chengdu 610041, China; shancuoji0131@163.com (S.);; 2Laboratory of Cardiac Structure and Function, Institute of Cardiovascular Diseases, West China Hospital, Sichuan University, Chengdu 610041, China

**Keywords:** pulmonary hypertension, mitral regurgitation, transcatheter edge-to-edge repair, transcatheter mitral valve repair, MitraClip

## Abstract

**Background:** Pulmonary hypertension (PH) is associated with the outcomes of mitral valve transcatheter edge-to-edge repair (M-TEER) in patients with severe mitral regurgitation (MR). However, the prognosis of baseline PH on MR patients after M-TEER has been controversial. This meta-analysis aimed to determine the prognostic value of PH with early and late outcomes after M-TEER with MitraClip. **Methods:** We systematically searched PubMed/MEDLINE, EMBASE, the Cochrane Central Register of Controlled Trials (CENTRAL) and Web of Science for studies. The results of the meta-analysis are summarized as the hazard ratio (HR), odds ratios (ORs) or mean difference (MD) and 95% confidence interval (CI). **Results:** A total of 20 publications were included in the systematic review, of which six were observational cohort studies including 5684 patients. The pooled incidence estimate of all-cause mortality was more common in severe PH than in patients who were non-PH. On pooled multivariate analysis, baseline PH was associated with late (≥1-year) all-cause mortality (HR = 1.61, 95% CI [1.23–2.11]) and the combined outcome of late HF rehospitalization and all-cause mortality (HR = 1.33, 95% CI [1.15–1.53]) after M-TEER. The level of SPAP significantly decreased after MitraClip in MR patients with PH (MD = −12.33 mmHg, 95% CI [−14.08–−10.58]). **Conclusions:** Baseline PH had a worse prognosis of early (≥30-day) cardiac mortality, late all-cause mortality as well as the composite outcome of HF rehospitalization and all-cause mortality after M-TEER compared to non-PH patients. Future studies are needed to prove these findings.

## 1. Introduction

Mitral regurgitation (MR) is the most prevalent valvular heart disease (VHD) requiring treatment with an adverse prognosis, which brings a heavy burden to public health and impairs quality of life [1]. Transcatheter mitral valve repair (TMVr) is the potential alternative therapy for high-surgical-risk patients suffering severe MR so far; in particular, mitral valve transcatheter edge-to-edge repair (M-TEER) is available for degenerative/primary MR (DMR) and functional/secondary MR (FMR) patients with a favorable physiology and anatomy whose symptoms persist despite with optimal guideline-directed medical therapy (GDMT) [2,3,4]. Pulmonary hypertension (PH) is a heterogenous disease entity presenting with different clinical phenotypes, which frequently occurs as the downstream consequence and not the cause of MR, and it is referred for surgical or catheter-based mitral valve replacement or repair for severe MR [5,6,7]. Previous studies have shown that PH has been associated with increased mortality and morbidity following the intervention for MR, and often marks a phenotype of disease progression [8,9,10,11]. Based on the pooled available literature, there are discordant hemodynamic definitions of PH in studies. The prognostic impact of PH, defined as an elevated systolic pulmonary artery pressure (SPAP) ≥ 50 mmHg assessed by Doppler echocardiography [12] or a mean pulmonary artery pressure (mPAP) ≥ 25 mmHg measured invasively by right heart catheterization (RHC) at rest [13], on sever FMR patients following M-TEER with the MitraClip system has not been established. Thus, our aims for this meta-analysis are to systematically review the studies and qualitatively and quantitatively synthesize the literature on the association of baseline PH and early and late outcomes after percutaneous mitral valve repair (PMVr) with MitraClip.

## 2. Methods

### 2.1. Search Strategy

According to the Preferred Reporting Items for Systematic Reviews and Meta-Analysis (PRISMA) guidelines, a systematic review protocol was registered with the International Prospective Register of Systematic Reviews PROSPERO (CRD42025643035), and a literature search was conducted across the PubMed/MEDLINE, EMBASE (Excerpta Media Database), the Cochrane Central Register of Controlled Trials (CENTRAL) and the Web of Science from inception to September 2024, based on the PICO (population, intervention or exposure, comparison, outcome) strategy using the following keywords: “mitral regurgitation”, “pulmonary hypertension”, “transcatheter mitral valve repair”, “edge-to-edge repair”, “percutaneous mitral valve repair”, “MitraClip”, “TMVr”, “TEER”, “PMVr”, “PH”, “MR” for English-language, peer-reviewed publications. The search code was used for all analyses in the review as presented in Appendix A. References for original and review articles were cross-checked. Institutional review board approval and patient consent were not required because of the systematic review and meta-analysis nature of this study.

### 2.2. Study Selection and Eligibility Criteria

Studies were included in the meta-analysis that fulfilled one of the following criteria: (1) in the English language and publication in peer-reviewed journals; (2) reporting PH after TMVr in MR; and (3) including quantitative raw data for predictors of interest, both with the availability of primary outcome data. Identified records were imported in EndNote X9.3.3. We followed the PRISMA checklist for the protocol of this meta-analysis. Studies were excluded that: (1) duplicated or overlapped publication; (2) constituted articles without primary data; (3) involved patients treated by MR surgery; (4) were non-human studies, conference abstracts, editorials or reviews.

### 2.3. Data Extraction and Outcomes

Two physician reviewers, SCJ and YBL, independently evaluated study eligibility and study quality and performed data extraction using standardized data collection sheets, and used the Newcastle–Ottawa quality scale (NOS), which evaluates in 3 domains, selection, comparability, and outcomes. The year of publication, study design, the number of enrolled centers, countries, the mean or median age of population, and the number of enrolled PH and non-PH patients were collected from each study. Discrepancies in the selection of relevant studies and data extraction were solved by consensus after consulting a third investigator, JLL. Data on study and patient characteristics, clinical and echocardiographic characteristics, and outcomes were extracted. For PH as a dichotomous parameter, HR and 95% CI were reported at 50 mmHg in SPAP or 25 mmHg in mPAP. Meanwhile, for PH as a continuous parameter, HR and 95% CI were reported per 1 mmHg in increased SPAP. The extracted information was tabulated according to pre-determined templates specified in our protocol (PROSPERO: CRD42025643035). Data extracted from the included studies and used for all analyses in the review are presented in Figure 1.

### 2.4. Statistical Analysisz

The results of the meta-analysis were summarized as hazard ratio (HR) or odds ratio (OR) and 95% CI. HR and OR were pooled using inverse variance weighting and Mantel–Haenszel weighting, respectively. Heterogeneity across studies was tested by the Cochran’s Q statistic and Higgins’ and Thompson’s I^2^ statistics. I^2^ > 50% and *p* ≤ 0.1 were considered to constitute significant heterogeneity, where random-effect models were used to estimate the between-study variance. Otherwise, the fixed-effect model was used for this meta-analysis. *p* < 0.05 was considered as statistically significant for other results. All analyses were conducted using Review Manager version 5.4 (The Cochrane Collaboration).

## 3. Results

### 3.1. Search Results and Study Quality Evaluation

A total of 941 studies were screened at title and abstract level, of which 20 [14,15,16,17,18,19,20,21,22,23,24,25,26,27,28,29,30,31,32,33] were examined in full text and ultimately included in this meta-analysis, considering that MitraClip approval was in 2008. The study selection process is shown in Figure 1. We pooled only six observational cohort studies [14,23,24,27,28,32] that reported early and late outcomes after M-TEER for patients with baseline PH vs. non-PH patients (encompassing 1163 patients with PH and 4523 without PH), and pooled HR for ≥1-year all-cause mortality after MitraClip as the primary endpoint of the analysis, which fulfilled the prespecified inclusion criteria for the present systematic review and meta-analysis (Table 1A). One prospective registry and one MitraClip arm from the Cardiovascular Outcomes Assessment of the MitraClip Percutaneous Therapy for Heart Failure Patients with Functional Mitral Regurgitation (COAPT) trial [24]. Most, 14 out of 20, eligible studies were single-center and retrospective in nature, some of which [15,16,17,18,19,20,21,22,25,26,29,30,31,33] were not specifically designed to assess the impact of PH on clinical outcomes, pooled univariate and multivariate HRs for ≥1-year all-cause mortality after TMVr (Table 1B). All studies were categorized as high quality, with NOS ranging from 6 to 8 (Appendix A).

### 3.2. Baseline Characteristics

Overall baseline characteristics are outlined in Table 1. The total number of enrolled patients in the individual studies ranged from 91 to 4071, accounting for 5684 patients. Patients had a mean age of 79.2 ± 8.9 years and were predominantly male (58.9%). The follow-up duration ranged from 1 year to 5 years.

### 3.3. Early and Late Outcomes and Sensitivity Analysis

#### 3.3.1. Pooled ORs of Early (30-Day) Cardiac and Non-Cardiac Mortality After M-TEER

In this quantitatively pooled analysis, the ≥30-day cardiac mortality as an outcome was used by three studies in total. Compared with non-PH patients, those with PH had a significant statistically difference in terms of early (≥30-day) cardiac mortality (OR 1.37, 95% CI [0.98–1.91], *p* = 0.06), without significant heterogeneity (I^2^ = 0, Q test *p =* 0.42). However, baseline severe PH had non-significant statistically difference in terms of ≥30-day non-cardiac mortality (OR 0.90, 95% CI [0.38–2.15], *p* = 0.82), with moderate heterogeneity (I^2^ = 57%, Q test *p =* 0.10) (Figure 2A). The random-effect model was used to combine the results.

#### 3.3.2. Pooled ORs of Early (30-Day) and Late (≥1-Year) All-Cause Mortality After PMVr

The three pooled studies reported the prognosis of PH on 30-day and ≥1-year mortality, amounting to a total number 794 and 3646 patients at 30-day follow-up, 614 and 2658 patients at 1-year follow-up, between patients with and without PH. A forest plot demonstrates a statistically significant difference between the pooled 30-day all-cause mortality (OR 1.62, 95% CI [1.20–2.20], *p =* 0.02) and the ≥1-year all-cause mortality (OR 1.57, 95% CI [1.28–1.93], *p* < 0.0001). As both of these demonstrated insignificant heterogeneity (I^2^ = 0, Q test *p* = 0.61, and I^2^ = 0, Q test *p* = 0.93, respectively) the fixed-effect model was used (Figure 2B).

#### 3.3.3. Echocardiographic Follow-Up: Pooled Estimated MD of SPAP After M-TEER Within 30 Days

In terms of the pooled pre- and post-MitraCip SPAP, there are three studies that reported this content. Overall, there was a significant improvement of SPAP elevation after the M-TEER at 30-day follow-up compared to baseline (MD = −12.33 mmHg, 95% CI [−14.08–−10.58]), and no heterogeneity was observed (I^2^  = 0, Q test *p* = 0.55), which was statistically significant (Z = 13.83, *p* < 0.00001) (Figure 3).

#### 3.3.4. Late (≥1-Year) HF Rehospitalization After TMVr (Univariate Analysis)

For the four included studies assessing SPAP as a dichotomous parameter, as the heterogeneity test (I^2^ = 63%, Q test *p* = 0.03) showed the considerable heterogeneity among these studies, the random-effect model was performed to the combined results. The prognosis on late (≥1-year) HF rehospitalization in patients with PH (univariate HR = 1.10, 95%CI [0.89–1.36]) indicated a non-significant difference (Z = 0.90, *p* = 0.37) (Figure 4).

#### 3.3.5. Late (≥1-Year) All-Cause Mortality After TMVr (Univariate Analysis and Multivariable Analysis)

When assessing SPAP as a continuous parameter, with a heterogeneity test (I^2^ = 44%, Q test *p* = 0.08) that indicated significant heterogeneity among these seven studies, the random-effect model was used to combine the results (Figure 5A). The pooled HR revealed that the baseline PH (elevated SPAP) was insignificantly associated with ≥1-year all-cause mortality (Univariate-HR = 1.01, 95% CI [1.00–1.02]), and that this was statistically significant (Z = 2.66, *p* = 0.08). A total of nine studies reported HR for the all-cause mortality evaluating elevated SPAP as a dichotomous variable; the pooled overall incidence of the late (≥1-year) all-cause mortality (multivariate HR = 1.61, 95% CI [1.23–2.11]) was statistically significant (Z = 3.48, *p* = 0.0005). However, the heterogeneity test [I^2^ = 60%, Q test *p* = 0.01] indicated significant heterogeneity, respectively, among these studies (Figure 5B).

#### 3.3.6. Late (≥1-Year) Combined Outcome of HF Rehospitalization and All-Cause Mortality After M-TEER (Multivariable Analysis)

Three studies with 4261 patients reported the HR for the combined outcome of HF rehospitalization and all-cause mortality. Since sensitivity analysis showed insignificant heterogeneity (I^2^ = 25%, Q test *p* = 0.5), the random-effect model was used (Figure 5C). The pooled results show that baseline PH was associated with a late (≥1-year) combined outcome (Multivariate-HR = 1.33, 95% CI [1.15–1.53]) after M-TEER, which was statistically significant (Z = 3.86, *p* = 0.0001).

### 3.4. Publication Bias

We only evaluated the publication bias of the pooled analysis of late all-cause mortality after M-TEER with MitraClip (both SPAP as a continuous and dichotomous parameter and mPAP as a dichotomous parameter) due to limited data in other pooled results. Visual examination of the funnel plot symmetry did not indicate publication bias in the included studies (Appendix A).

## 4. Discussion

### 4.1. Principal Findings

This was a meta-analysis of mainly retrospective observational studies, the main results of the meta-analysis were as follows: (1) patients with symptomatic severe FMR-concomitant PH had a higher risk for early or late all-cause mortality after M-TEER; (2) on pooled univariate and multivariate analysis, PH was associated with the pooled estimated of late (≥1-year) all-cause mortality (Multivariate-HR = 1.33, 95% CI [1.15–1.53]), the combined outcome of HF rehospitalization and all-cause mortality after PMVr (Multivariate-HR = 1.43, 95%CI [1.16–1.77]); (3) the level of SPAP significantly improved after M-TEER (MD = −12.33 mmHg, 95% CI [−14.08–−10.58]). Herein, baseline PH is associated with poor prognosis in FMR patients who underwent TMVr with MitraCip, even despite a substantial reduction in SPAP. Although prior studies have investigated the implications of PH (elevated SPAP) on TMVr, overall, these results are also intriguing. When assessing SPAP as a continuous parameter or a dichotomous variable, respectively, there are discordant pooled HRs for early and late all-cause mortality after M-TEER. Additionally, compared with non-PH patients, those with PH had a higher risk of early (≥30-day) cardiac mortality. These findings emphasize the heterogeneity of PH, and the clinical prognosis of catheter-based intervention in determining the extent to which elevated SPAP or clinical types of PH are directly or indirectly related to severe FMR.

### 4.2. Potential Mechanisms

PH is a common pathophysiological consequence of left-sided VHD, which is also a frequent complication of severe FMR due to flailing mitral valve leaflets and caused by alterations in left ventricle (LV) or left atrial (LA) function or geometry [6,34]. Indeed, the underlying pathogenesis of PH in chronic FMR typically involves a constellation of processes: in the initial compensated phases; pulmonary vascular resistance (PVR) is normal, and patients remain asymptomatic. A reversible transmission of elevated LA pressure to the pulmonary circulation constitutes a direct relationship. Chronic LA pressure/volume overload might irreversibly increase PVR and alter the pulmonary vascular bed over time; in a subset of PH-LHD patients this can result in right ventricle (RV) dilatation and hypertrophy in the final decompensated phase, which in turn might lead to tricuspid regurgitation, RV and/or LV enlargement or dysfunction, a vicious circle of MR, in an indirect relationship [35,36]. In those patient populations with severe FMR and PH, even though the precise factors that determine the compensation and decompensation stage changes in pulmonary vascular and right heart structure and function remain unclear, the absence of left-side cardiac pathology and elevated SPAP and/or mPAP may lead to a denial of the contribution of MR to PH-LHD [37,38]. Meanwhile, the hemodynamic characteristics and the prognostically relevant indexes of left-side heart function and clinical data regarding the short- or long-term evolution of pulmonary vascular and left heart alteration changes in severe FMR patients, have yet to be studied. Although RHC is the gold standard for diagnosing PH and its subtypes, echocardiography has become established as the primary tool for evaluating the diagnosis and severity of PH. In addition, in previous studies there has been a positive correlation between RHC and echocardiography measurements, since most PH pharmacotherapeutic intervention studies have utilized the classical definition of mean pulmonary arterial pressure (mPAP) at rest > 25 mmHg based on an RHC or SPAP ≥ 50 mmHg, assessed by echocardiography, while no prior clinical study in M-TEER has estimated both echocardiography and RHC measurements [39,40]. The 2020 American Heart Association/American College of Cardiology guidelines do not specially recommend an SPAP cut-off to guide mitral valve intervention [3]. However, in accordance with the updated 2021 guidelines of the European Society of Cardiology (ESC) and the European Respiratory Society (ERS), the current definition of PH is based on an mPAP at rest > 20 mmHg assessed by cardiac catheterization, and may further distinguish the hemodynamic PH subgroups and stage disease severity, or emphasize earlier diagnosis. Indeed, the World Health Organization (WHO) has classified PH into five clinical categories: pulmonary arterial hypertension; PH with left-side heart disease (PH-LHD); lung diseases and/or hypoxia; pulmonary artery obstructions (particularly thromboembolic PH); and undifferentiated or multifactorial causes. Among these, PH-LHD is the most prevalent subtype with an estimated prevalence of approximately 50% to 80%, which can be classified as pre-capillary PH or isolated post-capillary PH (Ipc-PH) and combined post- and pre-capillary PH (Cpc-PH) consistent with the general definitions of PH [3,6,41]. VHD-associated PH is considered a subgroup within WHO group 2. Typically, chronic PH-LHD secondary to MR development leads to persistent PVR and fibrosis and the remolding of pulmonary vascular via pulmonary arterial and arteriole vasoconstriction and thickening characterized by endothelial injury and/or dysfunction, resulting in excessive collagen deposition, an imbalance in the endothelial production of nitric oxide (NO) and endothelin-1 (ET-1) [42]. Several studies have demonstrated that NO and ET-1, as a mediators of endothelial dysfunction, have potential involvement in WHO group 2 PH (PH-LHD) [43,44]. Although pharmacotherapy with basic NO pathway-based pulmonary vasodilator-inhaled NO or phosphodiesterase 5 inhibitors (or any other pulmonary vasodilator, e.g., ET blockers or a prostacyclin analog/receptor) translates to real-world routine practice for PH, the clinical impact of ET blockers and NO in PH-LHD remains to be determined since most trials did not have positive results [45,46,47]. The current guidelines recommend phosphodiesterase 5 inhibitors for patients with Cpc-PH but not for Ipc-PH [6]. Two randomized, controlled, double-blind studies reveal that there was a positive outcome due to the use of sildenafil post procedure in PH patients following mitral valve intervention [48,49]. There are insufficient data to recommend the off-label use of pulmonary vasodilators in PH patients in secondary VHD (PH-LHD). Yet no approved pharmacological therapies exist that directly target PH-LHD. The latest ESC/ERS guideline recommends mitral valve surgery for patients with severe DMR-concomitant PH (defined as SPAP ≥ 50 mmHg) as carrying an acceptable surgical risk [12]. Meanwhile, the threshold of PASP at 50 mmHg as an indication for M-TEER has not been validated yet [50]. Importantly, given the threshold of PH as an indication for TMVr, but whether SPAP and/or mPAP should represent a sufficient parameter to guide M-TEER, or which PH-LHD patient subsets (both Ipc-PH and Cpc-PH) acquire clinical benefits and significant hemodynamics improvement after M-TEER, should be the focus of future studies. Given that most of the included studies did not distinguish between different types of PH (pre-capillary, IpcPH, and CpcPH) and different degrees of severity of PH (mild, moderate or severe PH), this study cannot systematically evaluate the interventional therapies in PH-LHD based on the currently available data. To summarize, so far, despite only a few prior data being available about the hemodynamic severity and types of PH in this meta-analysis, which showed the standard MD of SPAP before and after M-TEER within a 6-month follow-up, it seems that improvement in SPAP (pooled standard of MD = −1.03 mmHg) did occur in the severe PH group (Appendix A). A recent post hoc analysis reported a correlation between different hemodynamic types of PH and the hemodynamic response to MitraClip and the hemodynamic improvements in patients with Ipc-PH and Cpc-PH; however, the prognostic value of clinical outcomes is limited [33]. Therefore, these results need to be confirmed in larger future studies. Other invasive hemodynamic markers, such as pulmonary arterial compliance and right ventricular–pulmonary artery coupling have proven to be promising prognostic markers in cohort studies, but have not been tested in clinical trials. Additionally, it is important to distinguish PH from PH-LHD in isolation, and no prior study specializing M-TEER has evaluated both echocardiography and invasive measurements for PH. Therefore, the investigation of such markers and measurements would be of clinical interest to facilitate the identification of those FMR patients with significant pulmonary vascular disease who might possibly respond well to M-TEER therapy. A comprehensive evaluation of PH before a MitraClip intervention is necessary for selecting appropriate patients with FMR.

M-TEER with MitraClip mimicking the surgical Alfieri technique is a mature and guideline-recommended interventional method for high-surgical-risk patients with severe symptomatic MR [51]. The MitraClip pulls the anterior and posterior flailing leaflets together and creates a double-orifice mitral area, which modifies the physiology of diastolic transvalvular flow and reduces or eliminates MR severity, also potentially improving PH-LHD and LV afterload. Typically, after MitraClip use, the reversibility of PH depends on the severity and subtype of the FMR, and the pathophysiological adaptations of left heart. In this analysis, we observed an improvement in SPAP with a pooled estimate of MD = −12.33 mmHg beyond the 30-day post-M-TEER in patients with severe PH compared to baseline SPAP levels, indicating an early MR correction might improve severe PH. Prior studies have reported that patients with PH after TMVr would present a significant improvement of SPAP, which is consistent with the finding of this meta-analysis [52,53]. Similarly, patients with improved SPAP after TMVr could achieve better clinical outcomes when compared with those without a decrease in SPAP [27]. Previous studies have shown that LV unloading in patients who coexist with PH leads to a significant reduction in SPAP and transpulmonary gradient, highlighting the reversibility of PH [53,54]. It is reasonable to hypothesize that MitraClip treatment reduces the MR, resulting an improvement in the pulmonary vascular component, especially the post-capillary component, which decreases elevated SPAP/mPAP in PH patients, and consequently improves LV/LA volume and function, breaking the vicious circle that can lead to a worsening of PH-LHD. Accordingly, given the potential reversibility of PH, and the efficacy and safety profiles of MitraClip therapy, it is crucial to determine the optimal timing of M-TEER in patients with MR-concomitant PH or medically manage the pre-capillary PH in the pre-MitraClip phase, and define the role of PMVr in patients with PH-LHD. Furthermore, this meta-analysis showed that MR patients with PH are not only associated with ≥30-day cardiac mortality, but also with ≥1-year all-cause mortality and the composite outcome after M-TEER. In the two prior analyses, both COAPT and percutaneous repair with the MitraClip device for severe functional/secondary MR (MITRA-FR) trials demonstrated discordant results that might come from different inclusion criteria; it seems appropriate that patient selection would be needed for successful PMVr. In the COAPT trial, patients treated with M-TEER in combination with GDMT had a lower incidence of mortality and HF rehospitalization regardless of elevated SPAP (PASP > 50 mm Hg), which indicated a role for M-TEER in addition to GDMT in managing patients with severe FMR [2,24]. Conversely, in the MITRA-FR trial, enrolled patients who had a high LV dimension showed no clinical benefit from the MitraClip therapy over GDMT in patients with severe FMR [55]. The contemporary randomized study of the MitraClip device in heart failure patients with clinically significant functional MR (RESHAPE-HF2) is a prospective, randomized, multicenter trial including patients with moderate-to-severe FMR and with a higher proportion of patients on GDMT than in the other two studies, which reported that earlier MitraClip treatment could reduce HF hospitalizations and improve quality of life in these patients [56]. Moreover, patients in the MITRA-FR trial and RESHAPE-HF2 trial had a mean LV end-diastolic volume (EDV) of 252 mL and 211 mL, respectively, compared with the EDV of 194 mL observed in the COAPT trial. These results demonstrate that M-TEER improves outcomes in patients with FMR, and assessed the impact of LV remodeling on MitraClip treatment success. In clinical practice, clarifying which patients who have risk-to-benefit and cost-to-benefit ratios that favor catheter-based therapy and the selection criteria for M-TEER, especially in terms of identifying risk factors associated with poor outcomes after intervention and in FMR patients with PH, are critical. Future intervention studies are needed to systematically collect both invasive and non-invasive data to characterize PH (especially PH-LHD) in FMR to further identify the hemodynamic and clinical profile of the PH substrate, suggesting probable benefits or the lack thereof.

### 4.3. Limitations

Several limitations merit acknowledgement in this meta-analysis: (1) As all the included studies were observational and non-randomized trails, the quality of the published data is relatively low. (2) The eligible studies were relatively rare in terms of the establishment of prognostic values. In addition, we are unable to further analyze the impact of different clinical outcomes and post-procedural complications after TMVr due to these data not being reported fully in the primary studies. (3) As mentioned before, the cut-off for the definition of severe PH remains undetermined; here using 50 mmHg by non-invasive Doppler echocardiography as the cut-off value, whereas invasive RHC was not measured in the included studies. (4) Although data on HR for the combined outcomes and estimated MD for SPAP after M-TEER were included in only three studies, respectively, and sensitivity analysis was performed, we found that randomly deleting the articles in these studies did not affect the results of the study, indicating that the meta-analysis was robust. While meta-analyses can enhance the precision of event rate estimation, aggregating studies with diverse designs and varying quality may also introduce systematic bias. Owing to the aforesaid drawbacks, the aims of this review could not be accomplished fully.

## 5. Conclusions

The present study found that baseline SPAP might serve as a valuable parameter for the identification of patients at a high risk of severe FMR even after successful M-TEER. Despite MitraClip intervention improving the hemodynamics of PH, severe PH resulted in a worse prognosis in patients with MR compared to non-PH patients. Further and larger multi-center and adequately powered randomized trials with longer-term follow-up are needed to determine whether SPAP/mPAP is a sufficient parameter to guide M-TEER in FMR, and whether early intervention for MR-concomitant PH-LHD patients for whom evidence for the use of PH medications is limited might improve clinical outcomes, and which patient subsets will benefit the most from M-TEER with MitraClip.

## Figures and Tables

**Figure 1 diagnostics-15-00852-f001:**
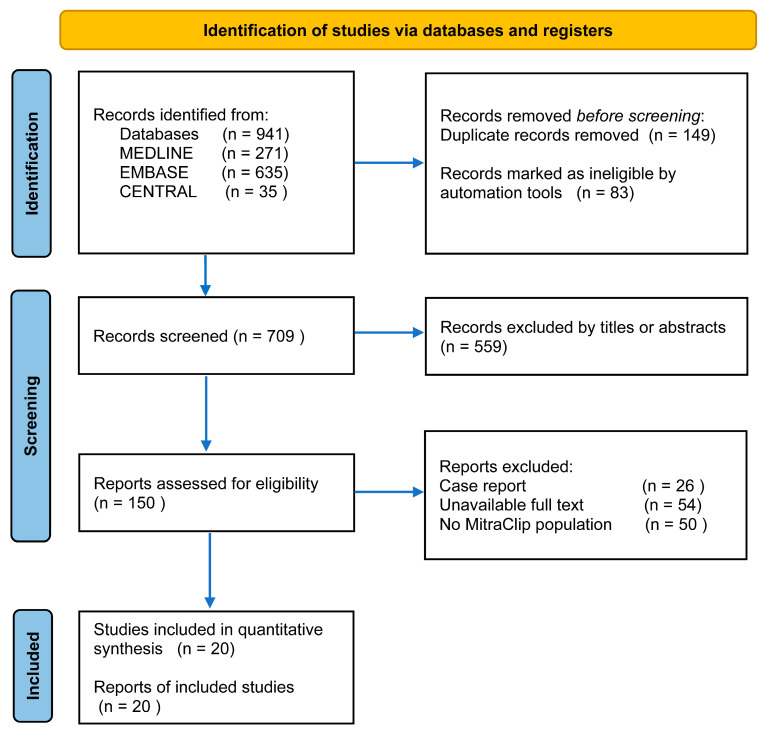
Study selection. Flow diagram based on the Preferred Reporting Items for Systematic Reviews and Meta-Analyses (PRISMA).

**Figure 2 diagnostics-15-00852-f002:**
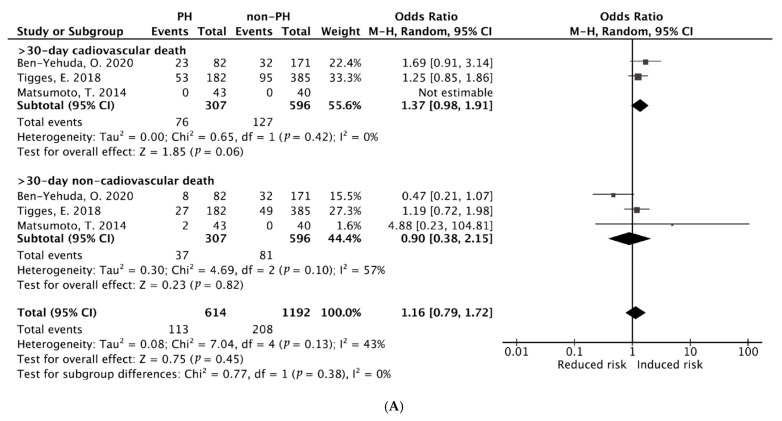
(**A**) Pooled ORs for early (≥30-day) cardiac and non-cardiac mortality. (**B**) Pooled ORs for early (30-day) and late (≥1-year) all-cause mortality [14,23,24,28].

**Figure 3 diagnostics-15-00852-f003:**
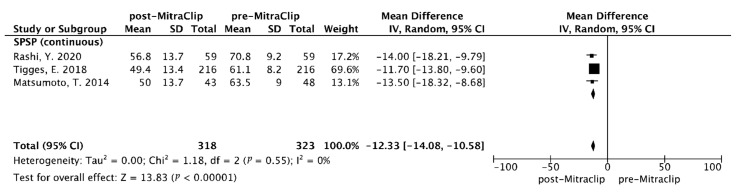
Forest plot summarizing mean difference (MD) for SPAP pre and post TMVr [14,27,28].

**Figure 4 diagnostics-15-00852-f004:**
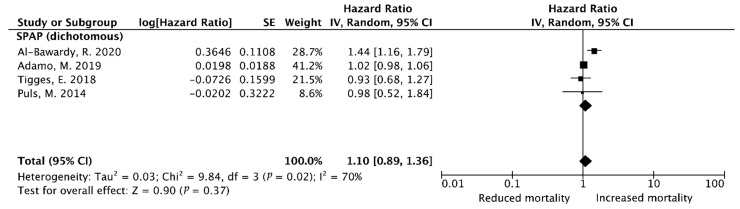
Pooled outcome of ≥1-year heart failure-related rehospitalization rates after TEER using MitraClip (univariate analysis) [15,20,23,28].

**Figure 5 diagnostics-15-00852-f005:**
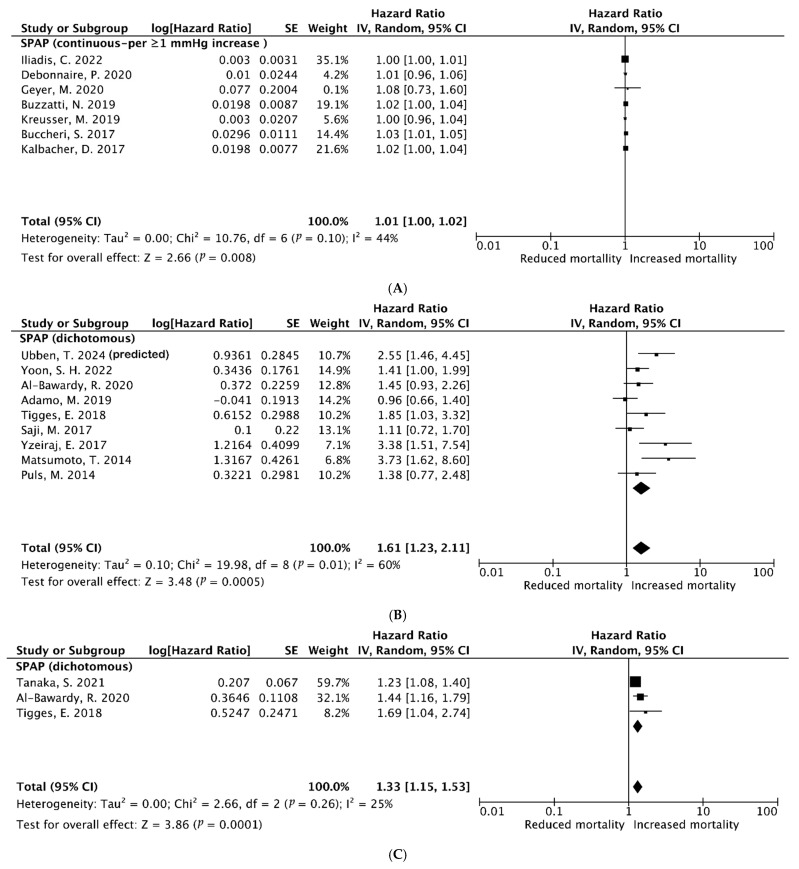
(**A**) Pooled hazard ratios (HRs) for late (≥1-year) all-cause mortality after TEER using MitraClip (Univariate analysis). (**B**) Pooled HR for late (≥1-year) all-cause mortality after TEER using MitraClip (multivariable analysis). (**C**) Pooled HR for the combined outcome of late (≥1-year) heart failure-related rehospitalization and all-cause mortality after TMVr using MitraClip (multivariable analysis) [14,15,16,17,18,19,20,21,22,23,25,26,28,29,30,31,32].

**Table 1 diagnostics-15-00852-t001:** Characteristics of the included studies.

**A. Pooled Characteristics of the Patients in 6 Cohort Studies**													
**First Author**	**Year**	**Region**	**Study Design**	**Sample Size**	**Baseline Characteristics**								
					**Age ± SD**	**Male (%)**	**No. of PH**	**MR Etiology (%)**	**SPAP ± SD**	**mPAP ± SD**	**Concomitant Lung/Cardiac Diseases**	**Follow-Up (Year)**	**Study Quality**
Ubben, T. [32]	2024	Germany	Retrospective cohort	449	76.0 ± 8.4	61.7	Yes: 200	PMR: 27.6	51.0 ± 14.0 (SPAPe)	25.0 ± 7.0	COPD (18.9) Stroke (10.6) Hypertension (79.9) AF (70.6) CAD (63.9)	2.8	7/8
							No: 249	FMR: 61.0					
Al-Bawardy, R. [23]	2020	United States	Retrospective cohort	4071	81.0 ± 8.0	57.7	Yes: 2968	PMR: 73.4	48.4 ± 17.1 (overall)	31.4 ± 11.1 (overall)	COPD (11.9) MI (24.7) Stroke (10.0) Hypertension (85.3) AF (62.0) CAD (6.9)	3.5	8/8
							No: 1103	FMR: 26.6					
Ben-Yehuda, O. [24]	2020	United States & Canada	Prospective cohort	253	72.9 ± 10.8	69.6	Yes: 82	PMR: 0.0	59.1 ± 8.8	NA	COPD (22.3) MI (53.8) Stroke (0.3) Hypertension (80.4) AF (56.0) CAD (53.8)	2	8/8
							No: 171	FMR: 100.0					
Rashi, Y [27].	2020	Israel	Retrospective cohort	177	74.8 ± 10.0	58.0	Yes: 59	PMR: 28.8	70.8 ± 9.2	NA	COPD (23.7) HF (98.3) AF (45.9) CAD (57.6)	1	8/8
							No: 118	FMR: 71.2					
Tigges, E. [28]	2018	Germany	Retrospective cohort	643	75.7 ± 8.4	59.8	Yes: 216	PMR: 31.4	61.1 ± 8.2	NA	COPD (23.6) MI (26.4) HF (21.8) Hypertension (78.7) CAD (76.4)	1	7/8
							No: 427	FMR: 68.6					
Matsumoto, T. [14]	2014	United States	Retrospective cohort	91	63.5 ± 9.0	61.5	Yes: 48	PMR: 0.0	63.5 ± 9.0	NA	COPD (18.7) HF (29.7) MI (31.9) Hypertension (80.2) AF (18.7) CAD (62.6)	3	8/8
							No: 43	FMR: 100.0					
**B. Pooled characteristics of the patients in other 13 studies.**													
**First Author**	**Year**	**Region**	**Study Design**	**Sample Size**	**Baseline Charateristics**								
					**All-Cause Death, n (%)**			**Age ± SD**	**Male (%)**	**Concomitant Lung/Cardiac Diseases (%)**	**MR Etiology (%)**	**Follow-Up (Year)**	**Study Quality**
Yoon, S. H. [30]	2022	United States & Netherlands	Retrospective cohort	380	2-year: 131 (31.0)			71.0 ± 13.0	61.1	COPD (6.1) MI (29.2) Stroke (7.1) Hypertension (82.9) AF (48.9)	DMR: 0	3	7/8
											FMR: 100		
Iliadis, C. [29]	2022	Germany	Retrospective cohort	1074	2-year: 289 (36.0)			75.0 ± 5.0	66.0	COPD (18) MI (39) Stroke (9) AF (59)	DMR: 0	5	6/8
											FMR: 100		
Tanaka, S. [31]	2021	Japan	Retrospective cohort	25	1-year: 4 (15.4)			87.4 ± 11.0	56.0	AF (36)	DMR: 16.0	1	6/8
											FMR: 84.0		
Debonnaire, P. [25]	2020	Belgium	Prospective cohort	107	1.5-year: 49 (19.0)			73.0 ± 10.0	70.1	CAD (77) MI (61) Stroke (12) AF (44)	DMR: 0	3	7/8
											FMR: 100		
Geyer, M. [26]	2020	Germany	Retrospective cohort	461	1.5-year: 112 (24.3)			78.6 ± 7.3	53.0	COPD (12.9) MI (22.9) Stroke (10.6) Hypertension (81.9) AF (48.9) CAD (60.9)	DMR: 42.6	1	8/8
											FMR: 57.4		
Kreusser, M. [22]	2019	Germany	Retrospective cohort	174	year:31 (17.8)			75.2 ± 11.9	69.5	COPD (19) Stroke (13.8) Hypertension (56.9) AF (36.2) CAD (57.5)	DMR:20.1	1	7/8
											FMR: 79.9		
Adamo, M. [20]	2019	Italy	Prospective cohort	304	Year: (15.1) 3-year: (35.5) 5-year: (47.3)			72.0 ± 10.0	63.8	COPD (21.7) HF (65.8) Hypertension (67.4) AF (41.2) CAD (54.3)	DMR:22.5	5	7/8
											FMR: 77.5		
Buzzatti, N. [21]	2019	Italy	Prospective cohort	339	5-year: (53.5)			72.0 ± 10.3	74.0	COPD (30.9) HF (11.9) Hypertension (81.9) AF (39.7) CAD (57.5)	DMR:27.5	5	7/8
											FMR: 68.6		
Kalbacher, D. [17]	2017	Germany	Retrospective cohort	766	year:154 (20.1)			75.3 ± 8.5	61.0	COPD (26.4) MI (26.4) Stroke (3.8) Hypertension (79.8) AF (55.7) HF (62.3)	DMR:29.8	1	8/8
											FMR: 70.2		
Buccheri, S. [16]	2017	Italy	Prospective cohort	311	Year: (16.5)			72.6 ± 9.9	59.8	COPD (21.9) Stroke (10.6) Hypertension (78.1) AF (40.5)	DMR:22.2	1	7/8
											FMR: 77.8		
Saji, M. [18]	2017	United States	Retrospective cohort	222	Year: 36 (15.1) 3-year: 38 (35.5) 5-year:33 (47.3)			77. ± 11.1	49.5	COPD (31.9) MI (31.5) Stroke (16.2) Hypertension (77.0) AF (62.1) CAD (62.1)	DMR: 69.4	5	7/8
											FMR: 30.6		
Yzeiraj, E. [19]	2017	Germany	Retrospective cohort	139	1.5-Year: 35 (25.2)			76.4 ± 7.5	61.2	COPD (17.3) MI (45.3) Stroke (10.1) Hypertension (81.9) AF (67.6) CAD (61.2)	DMR: 15.8	2	7/8
											FMR: 84.2		
Puls, M. [15]	2014	Germany	Retrospective cohort	150	1-Year:(21.5)			74.4 ± 9.3	63.0	COPD (22.0) Stroke (8.0) Hypertension (81.9) AF (66.0) HF (45.0)	DMR: 35.0	1	7/8
											FMR: 65.0		

AF: atrial fibrillation; CAD: coronary artery disease; COPD: chronic obstructive pulmonary disease; FMR: functional mitral regurgitation; DMR: degenerative mitral regurgitation; HF: heart failure; MI: myocardial infarction; MR: mitral regurgitation.

## Data Availability

The authors confirm that the data supporting the findings of this study are available within the article.

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
