# Peer review of "The Prognostic Value of Pulmonary Hypertension in Patients with Mitral Regurgitation Undergoing Mitral Valve Transcatheter Edge-to-Edge Repair: A Systematic Review and Meta-Analysis"

_diagnostics, 2025, doi:10.3390/diagnostics15070852_

Round 1
Reviewer 1 Report
Comments and Suggestions for Authors
This was a meta-analysis of mainly retrospective observational studies, the main results of the meta-analysis were as follows: Patients with symptomatic severe FMR concomitant PH had higher risk for early or late all-cause mortality after M-TEER; (2) On pooled univariate and multivariate analysis: PH was associated with the pooled estimated of late (≥ 1-year) all-cause mortality (Multivariate-HR=1.33, 95% CI [1.15-1.53]), the combined outcome of HF rehospitalization and all-cause mortality after PMVr (Multivariate[1]HR=1.43, 95%CI [1.16-1.77]); (3) The level of SPAP significantly improved after M-TEER (MD=-12.33 mmHg, 95% CI [-14.08– -10.58]).
Baseline PH have associated with poor prognosis in FMR patients who underwent TMVr with MitraCip, even despite a substantial reduction of SPAP
Pulmonary hypertension is associated with increased mortality and heart failure readmissions in patients undergoing TMVr using MitraClip.
These were conclusions of a large study by Evin Yucel and colleagues, on Curr Treat Options Cardiovasc Medicine 2019 Sep 10;21(10):60.” Pulmonary Hypertension in Patients Eligible for Transcatheter Mitral Valve Repair: Prognostic Impact and Clinical Implications".
In another study, ADITYA PATEL and colleagues showed that patients with PH who underwent TMVR with MitraClip had poorer in-hospital outcomes compared to those without PH. Vascular complications, major bleeding requiring transfusion and AKI were more prevalent in patients with PH post-procedurally. This highlights the importance of close monitoring and meticulous post-procedural management among patients with PH.
The study is interesting and well conducted but the results are already known in the literature and there are no major aspects of novelty, which should be found through the evaluation of pulmonary ventricular arterial coupling by example: TAPSE/PAPS ratio and sPAP/PAAT ratio. In this version the work cannot be published and I point out some articles published on the subject : 1) Right ventricular-pulmonary artery coupling and clinical outcomes after mitral transcatheter edge-to-edge repair: A meta-analysis P. Theofilis1 , E. Mantzouranis1 , P.K. Vlachakis1 ,et al…..European Heart Journal (2024) 45 (Suppl 1)2) Potential Use of Systolic Pulmonary Artery Pressure/Pulmonary Artery Acceleration Time Ratio in Severe Functional Tricuspid Regurgitation with Pulmonary Hypertension..Walter Serra et al Cardiology 2024 Sep 30:1-7. doi: 10.1159/000541529
Reviewer 2 Report
Comments and Suggestions for Authors
I should congratulate authors for this manuscript. But they need to give more data for completion.
1) The study does not differentiate between primary and secondary Mitral regurgitation cases undergoing mitral procedure.
2) Role of other Echo parameters like LV Volume, LA Size, LV & RV function, TR etc are not mentioned.
3) We are not sure whether other causes of PAH were excluded from the study.
4) There is no mention of cath data during index procedure.
5) The response to medications are not mentioned.
6) Presence of concomitant cardiac diseases is not mentioned.
I think these will make your manuscript complete.
